# Relationship between the Ubiquitin–Proteasome System and Autophagy in Colorectal Cancer Tissue

**DOI:** 10.3390/biomedicines11113011

**Published:** 2023-11-09

**Authors:** Martyna Bednarczyk, Małgorzata Muc-Wierzgoń, Sylwia Dzięgielewska-Gęsiak, Dariusz Waniczek

**Affiliations:** 1Department of Hematology and Cancer Prevention, Medical University of Silesia in Katowice, 40-055 Katowice, Poland; mbednarczyk@sum.edu.pl; 2Department of Preventive Medicine, Medical University of Silesia in Katowice, 40-055 Katowice, Poland; sgesiak@sum.edu.pl; 3Department of Surgical Nursing and Propaedeutics of Surgery, Medical University of Silesia in Katowice, 40-055 Katowice, Poland; dwaniczek@sum.edu.pl

**Keywords:** ubiquitin–proteasome system, autophagy, colorectal cancer tissue

## Abstract

Background: Dysregulation of the autophagy process via ubiquitin is associated with the occurrence of a number of diseases, including cancer. The present study analyzed the changes in the transcriptional activity of autophagy-related genes and the ubiquitination process (UPS) in colorectal cancer tissue. (2) Methods: The process of measuring the transcriptional activity of autophagy-related genes was analyzed by comparing colorectal cancer samples from four clinical stages I-IV (CS I-IV) of adenocarcinoma to the control (C). The transcriptional activity of genes associated with the UPS pathway was determined via the microarray technique (HG-U133A, Affymetrix). (3) Results: Of the selected genes, only *PTEN-induced kinase 1 (PINK1)* indicated statistical significance for all groups of colon cancer tissue transcriptome compared to the control. The transcriptional activity of the *protein tyrosine phosphatase non-receptor type 22 (PTPN22*) gene increased in all stages of the cancer, but the *p*-value was only less than 0.05 in CSIV vs. C. *Forkhead box O1 (FOXO 1*) and *ubiquitin B (UBB)* are statistically overexpressed in CSI. (4) Conclusions: The pathological expression changes in the studied proteins observed especially in the early stages of colorectal cancer suggest that the dysregulation of ubiquitination and autophagy processes occur during early neoplastic transformation. Stopping or slowing down the processes of removal of damaged proteins and their accumulation may contribute to tumor progression and poor prognosis.

## 1. Introduction

Colorectal cancer is the third most common malignant tumor in the world and one of the leading causes of oncological mortality. Although many factors have been known to affect cancer, we still do not know the essential details of this complex pathogenesis. In recent years, particular attention has been paid to the apoptotic imbalance of tumor cells compared to healthy tissue, which can influence the uncontrolled growth of the tumor cell population. Every day, circa 3–5% of intracellular proteins are removed using lysosomal or ubiquitous mechanisms, which can activate autophagy and programmed cell death [1,2]. Autophagy is activated in response to long-term nutrient deficiency, tissue remodeling, organelle quality control, immune system responses, and cellular stress [3,4]. Damaged or as-synthesized proteins are degraded by the ubiquitin–proteasome system (UPS). The ubiquitin–proteasome system is responsible for the degradation of damaged or as-synthesized proteins. In contrast to the autophagy–lysosome system, the UPS system is characterized by high selectivity because it only removes ubiquitin-labeled proteins [4]. 

The UPS degradation pathway consists of five steps, whereby the substrate with a polyubiquitin tail after recognition by 26S proteasome is subsequently degraded [5]. Currently, research into the association of ubiquitination with autophagy is underway. The authors indicate that the autophagy receptor p62/sequestosome-1 is the principal molecule that regulates the cross-talk between the two systems [5,6,7]. Ubiquitin is likely to play a role in selective autophagy and may also serve as a generic identifier for substrates [8]. It is interesting that with the chance of failure of the UPS system, autophagy is a compensatory mechanism in cells, and proteins involved in ubiquitination can eliminate UPS defects by increasing autophagy [9]. Recent studies indicate that the dysregulation of the ubiquitin-mediated system plays a crucial role in the pathogenesis of various autophagy-related human diseases, such as cardiovascular diseases, neurodegenerative diseases, infectious diseases, myopathies, metabolic syndromes, and tumorigenesis [2,3,7,8].

The present study analyzed changes in the transcriptional activity of autophagy-related genes and the ubiquitination process in colorectal cancer tissue, postulating their possible role in the disturbance of protein homeostasis and the progression of cancer. 

## 2. Materials and Methods

### 2.1. Materials

The 30 pairs of surgically removed tumor and healthy specimens from colorectal cancer (CRC) patients at clinical stages I–IV (CSI-CSIV) according to the 7th edition of the Union for International Cancer Control/American Joint Committee on Cancer were analyzed—see Table 1. Specimens were obtained during surgical resection of the colon affected by cancer. Healthy control specimens (C) were collected from an area 5 cm outside of the histologically negative margin [3]. 

The project was approved by the Bioethics Committee of the Medical University of Silesia in Katowice, decision no. KNW/0022/KB1/42/14.

### 2.2. Methods

The specimens were homogenized using POLYTRON® (Kinematics, AG, Flaach, Switzerland) and then the total RNA was isolated using TRIzol® reagent (Invitrogen Life Technologies, Carlsbad, CA, USA) according to the manufacturer’s instructions. The isolated RNA was purified using an RNeasy Mini Kit (Qiagen, Hilden, Germany) in combination with DNase I digestion. A Gene Quant II spectrophotometer was used to quantify the RNA concentration, based on absorbance at 260 nm. The transcriptional activity of genes associated with the UPS pathway was determined via the microarray technique (HG—U133A- Affymetrix Human Genome U133A Array, Affymetrix, Ltd., Santa Clara, CA, USA).

Results were analyzed using PL-Grid Infrastructure [10] and the Statistica 12.0 program. In the statistical analysis, a statistical significance level of *p* < 0.05 was used.

## 3. Results

Based on the Affymetrix database and the literature data, 1095 ID mRNAs of genes involved in degradation in the ubiquitin–proteasome system were selected from 22,283 mRNAs.

In the first stage of transcriptome analysis, the fluorescence distribution of mRNA fluorescent genes related to ubiquitination in particular groups was presented (Figure 1). 

The obtained results suggest the similarity of transcriptomes at the median level, whereas there are noticeable differences in the values of the quartile range and remote values depending on the CS.

The next step was to conduct a one-way ANOVA test using Benjamini–Hochberg correction and a post hoc Tukey test, where the task was to simultaneously compare all analyzed transcriptome groups with the control sample. The study found that among the 1095 mRNA groups of ubiquitin-associated genes, there were statistically significant differences (*p* < 0.05) in the fluorescence intensity of mRNA 180 ID. The detailed results of such an analysis are available in our previous article [2].

Next, the Panther program was used to carry out the overrepresentation test (with Bonferroni correction for mRNAs differentiating between the compared transcriptome groups, indicating biological processes, molecular functions, and ubiquitination signaling pathways). The directions of the observed changes were determined as follows: a reduction in gene expression (reduction in fluorescence signal) and an increase in expression, as well as an increase in fluorescence signal for the examined transcript. To assess the degree of fluorescence, the fold change (FC) parameter was used, which is the log2 value of the multiple of the signal difference. Based on the directions of the observed changes, and depending on the clinical stage of cancer, four genes were selected whose transcriptional activity underwent significant modifications—see Table 2.

The study showed that the majority of genes selected in the Panther program were reduced in all stages of colorectal cancer. Of the selected genes, PINK1 differentiates all groups of colon cancer tissue transcriptome from the control group. The exception is PTPN22, which shows increased transcriptional activity in all sections of the cancer intestine. Moreover, HSPA8 is overexpressed in CSI and CSII, and UBB is overexpressed in CSI. Simultaneously, HSPA8 mRNA levels were compared between CSI and CSIV and CSII and CSIII, because these values were statistically significant (*p* < 0.05).

## 4. Discussion

Cancer cell survival depends on coordinated anabolic and catabolic agents. Many proteins and organelles are cytotoxic if accumulated through uncontrolled action. Things get even more complicated when the cell is exposed to environmental stress. For cancer cells, these stresses will be hypoxia, lack of nutrients, or adjuvant chemotherapy during tumor treatment [11,12].

Two main reasons for protein degradation in cells are UPS and autophagy. In recent years, an increasing number of studies have indicated a coordinated and complementary relationship between these processes [13,14]. The molecular proteasome pathway, mediated by chaperones, is unable to degrade misfolded proteins. Subsequently, proteasome inactivation may occur, which results in cytotoxicity. Therefore, for the mediation of the degradation of ubiquitinated protein aggregates, the lysosomal autophagy pathway works as a compensatory mechanism [15].

The unfolded protein response (UPR) plays a significant role in the process of autophagy induced by proteasome inhibitors. During this process, proteasome is destroyed, which stops the degradation of the misfolded proteins and the formation of aggregates that will disturb endoplasmic reticulum homeostasis and induce UPR, e.g., a proteasome inhibitor activates autophagy through the inositol-requiring endonuclease–c-Jun N-terminal kinase/bcl-2 protein/Beclin-1 (IRE/JNK/bcl2/BECN1) axis [16,17].

As an additional feature, proteasome inhibitors are able to affect autophagy-related proteins directly to promote the autophagy process [15], e.g., a proteasome inhibitor disrupts the interaction between mammalian targets of rapamycin complex 1 -mTORC1′s (Raptor) structural components and its interacting partners, acting as an inhibitor of mTORC1 activity [18,19]. Also, after proteasome inhibition, tumor protein 53 (p53) accumulates and translocates into the nucleus, where it acts as a factor of transcription for genes such as DNA Damage Regulated Autophagy Modulator 1 (*DRAM1*) [20]. An increase in p53 inhibits the mTORC1 pathway and may activate the autophagy process [18,21]. In addition, a recent study found that baby microtubule-associated protein 1A/1B-light chain 3 (LC3) levels increase when something potentially cancerous is created as an effect of proteasome inhibitor activity [17,22].

During carcinogenesis, cancer cells are at risk of deficiencies of nutrients, an accumulation of acid metabolism products, hypoxia, a change in the number of chromosomes, the activation of oncogenes, and the inactivation of tumor suppressor genes. Because of which, the protein quality control (PQC) mechanism of the endoplasmic reticulum is activated. Primarily, the three UPR signaling pathways will block tumor growth in the early stages of cancer, while cancer cells will adapt to internal and external stress and resist apoptosis in later stages [23].

Importantly, any changes induced in the UPS system can potentially alter cellular homeostasis, and it was shown in the studies that proteasome activity is elevated in many human cancers. This may be caused by the fact that the key proteins are regulated by UPS in various cellular processes such as the epithelial–mesenchymal transition (EMT), the cell cycle, signal transduction, gene expression, DNA repair, and apoptosis [24].

Studies also report that the UPS process and autophagy are closely related, which is also indicated by the results of our research, where it can be observed that the studied genes participate in the processes of ubiquitination and autophagy. The UPS is involved in the degradation of unwanted and useless proteins in cells. Autophagy is used to massively degrade damaged cellular components and organelles, for the purpose of producing new building blocks such as amino acids, nucleotides, and sugars, which are essential for cell survival.

UPS and autophagy also act as post-translational regulators of biochemical pathways through the selection and mass degradation of damaged proteins, respectively [25,26]. Due to the diverse, complementary, and somewhat overlapping functions of these two mechanisms, there is a rich network of regulatory links between them that together exert a significant influence on stress responses and cell survival [27,28].

Therefore, when considering the modulation or inhibition of UPS or autophagy to design new treatments for colorectal cancer, we may find it helpful to take the interrelationships of these two degradation mechanisms that occur in CRC into consideration.

Based on the preliminary results presented in the above study, we can assume that the autophagy of ubiquitin-labeled substrates may be disturbed in colorectal adenocarcinoma.

In this study, our team selected several genes whose protein products play a role in the development of colorectal cancer. The most important of these appears to be the PTEN-induced kinase 1 (PINK1) protein, which differentiates all groups of transcripts from the control, showing a decrease in expression in all clinical stages. PINK1 is a serine-threonine protein mitochondrial kinase. It is a key protein involved in the degradation of damaged mitochondria (mitophagy) [5,8,9,29].

*PINK1* is translated into the cytoplasm and is transported across the outer mitochondrial membrane (OMM) and the inner mitochondrial membrane (IMM). It is then cleaved by mitochondrial processing peptidase (MPP) and presenilin-associated rhomboid-like protease (PARL), and then returns to the cytoplasm where it is normally degraded by the proteasome, inhibiting mitophagy through interactions with parkin and protein kinase A (PKA) [30]. *PINK1* is a potential suppressor of many cancers, including, for example, glioblastoma multiforme, and its overexpression may be a therapeutic strategy to inhibit the growth of cancer cells.

Under physiological conditions, the level of *PINK1* in mitochondria is very low, while during the depolarization of mitochondria, as an oxidative stress response, the *PINK1* protein accumulates in damaged organelles, crosses the outer mitochondrial membrane, and acts as a cellular indicator of damaged mitochondria, initiating autophagy [2,3].

In the studies performed so far, there is usually a decrease in *PINK1* expression in tumor cells compared to controls, which is consistent with our results [31,32]. Moreover, in some types of cancer, e.g., in lung cancer, *PINK1* expression is associated with poor prognosis and resistance to chemotherapy [33,34]. In addition, the action of *PINK1* primarily depends on the type of cancer and the stage of advancement. The literature data suggest that in the case of breast cancer, *PINK1* exhibits anticancer activity, while in colorectal cancer it is responsible for the proliferation of cancer cells [35].

The membrane-anchored mitofusion 1 (*MFN1*) protein is also involved in the process of mitophagy. Damaged proteins in the outer mitochondrial membrane can be selectively removed by the ubiquitin proteasomal system [36]. The newest research points out that the UPS is responsible for monitoring the import of the mitochondria protein and helps remove misplaced proteins inside the mitochondria [37]. Eventually, irreversibly damaged mitochondria that cannot be repaired can be completely eliminated by mitophagy [38]. Damaged mitochondria first cleave in a dynamin-related protein 1 (*DRP1*)-dependent manner. Mitochondrial fusion and fission processes are mediated by mitochondria-associated GTP-ases of the dynamin family. The fusion between OMMs is mediated by the MFN1 and MFN2 proteins and cleaved by the DRP1 protein. Both processes contribute to the maintenance of oxidative phosphorylation under stress conditions and the elimination of mitochondria damaged by oxidative stress [39,40].

To maintain cellular homeostasis under stressful conditions, both the UPS and autophagy pathways show high coordination and complement each other [41,42]. Under stressful conditions, specific proteins on the OMM selectively undergo ubiquitination followed by proteasome-mediated degradation in a process called mitochondrial-associated degradation (MAD) [43]. Mitochondrial depolarization results in the activation of *PINK1* in the OMM, which in turn summons the Parkin protein to damage mitochondria. Parkin is a 465-amino acid residue E3 ubiquitin ligase, which plays a critical role in ubiquitination.

Parkin recognizes proteins on the OMM upon cellular injury and mediates in the process of removing damaged mitochondria via autophagy and proteasomal mechanisms, while *PINK1* phosphorylates polyubiquitin chains, which leads to an amplification of the feedback signal. Polyubiquitin chains are recognized by a number of ubiquitin-binding proteins involved in autophagy [44,45]. In conclusion, oxidative stress was proven to trigger extensive mitochondrial ubiquitination and fragmentation, resulting in mitophagy.

In our study, we showed that *MFN1* expression is downregulated in CRC compared to controls. Moreover, the higher the stage of clinical advancement, the lower the transcriptional activity. This is also in line with other studies, e.g., similar results were obtained in lung cancer or breast cancer [46,47]. Decreased *MFN1* expression is possibly an indicator of poor prognosis and metastases [46,47,48].

The next selected gene was *Heat Shock Protein Family A (Hsp70)* Member 8 (*HSPA8*), because it is overexpressed in CSI and CSII and underexpressed in CSIII and CIV. HSPA8 belongs to the *HSP70* family of heat shock proteins and participates in chaperone-dependent aerophagy [49]. *HSPA8* shows overexpression in a variety of malignant tumor cells, which proves that it plays an important role during the creation and growth of these cells [50]. Moreover, reduced expression of *HSPA8* may act as an inhibitor for solid tumor cell growth, apoptosis inductor, and cell cycle arrest factor [51]. Studies have shown that *HSPA8* shows high expression in various cancer cells, such as hepatocellular carcinoma, thyroid cancer, cholangiocarcinoma, endometrial cancer, and colon adenocarcinoma, as it participates during cancer cell growth and regulates their autophagy [52,53,54,55,56]. Yang et al. showed that during early hepatocellular carcinoma, *HSPA8* has higher transcriptional activity in the tumor tissue, and this is correlated with poor patient prognosis [57]. This is also reflected in our study results for colorectal cancer. High *HSPA8* expression is associated with cancer-associated genomic changes that arise in the first stages of the disease, including the regulation of oncogenes and tumor suppressors [58].

The next selected gene was *ubiquitin B (UBB)*, which participates in the ubiquitination process. Its transcriptional activity was only higher than in controls in CSI. In turn, the UBB gene encodes polyubiquitin, conditioning the maintenance of ubiquitin homeostasis in the cell. Ubiquitin provided by this protein is used to label substrates destined for proteasome degradation [59,60,61]. Increased expression of *UBB*s has been shown in several tumors, indicating the maintenance of a high rate of tumor cell proliferation and support for their ability to overcome increasing cellular stresses [62,63,64]. In turn, *UBB* silencing in neuroblastoma, liver cancer, breast cancer, and prostate cancer cells significantly reduced the proliferation rate of cancer cells [65,66].

We observed a decrease in *UBB* gene activity for all clinical-stage cancers except CSI. However, since only three samples were included in this group, it is difficult to draw conclusions on the clinical relevance of early cancer observation. Similar observations concern the ++++ protein, which is a cytoplasmic chaperone, participating in the degradation pathways of damaged proteins in a chaperone-dependent autophagy [11]. The decline in transcriptional gene activity for *UBB* and *HSPA8* together with the clinical progression of colorectal cancer implies that cellular repair mechanisms may be undermined and may be both a symptom of progressive pathology and contribute to its further development.

The last gene identified was protein tyrosine phosphatase non-receptor type 22 (*PTPN22*). It was chosen because it is the only gene that is overexpressed in all stages of clinical cancer. The importance of *PTPN22* in autophagy has not yet been defined. However, it was found that it participates, among other things, in the development of inflammatory bowel diseases by affecting immunological processes, including the activation of T lymphocytes [67]. In patients with active Crohn’s disease, the observed change in *PTPN22* expression influenced the increase in proinflammatory cytokines Interleukin-6 (IL-6) and Interleukin-17 (IL-17), as well as the progression of the disease [68,69]. In our observation, elevated levels of *PTPN22* gene expression were noted, with the highest for the CSIV group, significantly differentiating them from the control.

An indication of the expression profiles of the autophagy- and ubiquitination-related genes allows us to further use them in prognosis [70]. After determining the correlation between transcriptional activity in these genes and disease activity parameters during different clinical stages, it is possible to use it as an advancement determination factor. Moreover, it has been described that genetic or pharmacological expression modulators for the studied genes may have therapeutic potential in the treatment of solid tumors. The combination of cell-death-related gene modulators with other drugs used in the treatment of colorectal cancer allows one to achieve a treatment response and overcome drug resistance.

Our study is a preliminary one. Despite obvious limitations, the confirmed results are the basis for continuing this study. Primarily, we are planning to continue the study using a larger number of patients and we plan to determine the concentration of proteins related to autophagy and ubiquitination using the ELISA method.

The authors believe that it is urgent and necessary to identify new cancer biomarkers and understand the molecular mechanisms involved in the formation and progression of cancer. This will allow the development of more effective diagnostic methods and treatment strategies.

## 5. Conclusions

Expression changes in the studied proteins observed especially in the early stages of colorectal cancer development suggest that the dysregulation of ubiquitination and autophagy processes occur during early neoplastic transformation. Stopping or slowing down the processes of the removal of damaged proteins and their accumulation may contribute to tumor progression and poor prognosis; this demands more studies in the future with the aim of improving cancer diagnosis and therapy.

## Figures and Tables

**Figure 1 biomedicines-11-03011-f001:**
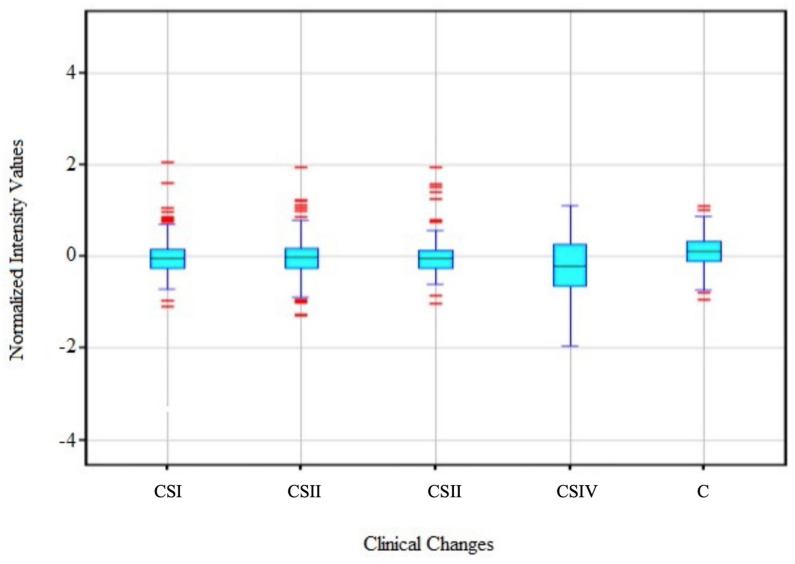
Distribution of mRNA fluorescence signals in study groups depending on the clinical stages (CSI−CSIV). Reprinted from [2] with permission from Bentham.

**Table 1 biomedicines-11-03011-t001:** Number of specimens in the study and control group.

	CSI	CSII	CSIII	CSIV	Control
Number of specimens	7	7	8	8	17

**Table 2 biomedicines-11-03011-t002:** Statistical significance of mRNA genes associated with protein ubiquitination and autophagy determined from over-representative test, where *p* < 0.05. Reprinted from [2] with permission from Bentham.

Affymetrix ID	Gene Name	*p*	Fold Change				Strongest Statistical Comparison
			CSI vs. C	CSII vs. C	CSIII vs. C	CSIV vs. C	
209018_s_at	*PINK1*	6 × 10^−6^	−2.12 ↓	−1.83 ↓	−1.40 ↓	−2.15 ↓	CSI vs. C CSII vs. C CSIII vs. C CSIV vs. C
201155_s_at	*MFN1*	2.7 × 10^−6^	−1.53 ↓	−1.69 ↓	−1.75 ↓	−2.39 ↓	CSIII vs. C
221891_x_at	*HSPA8*	2 × 10^−3^	1.19 ↑	1.16 ↑	−1.37 ↓	−1.37 ↓	CSI vs. CSIV CSII vs. CSIII
208011_at	*PTPN22*	4.9 × 10^−3^	1.06 ↑	1.01 ↑	1.07 ↑	1.18 ↑	CSIV vs. C
208945_s_at	*BECN1*	8.9 × 10^−7^	−1.32 ↓	−1.42 ↓	−1.63 ↓	−1.77 ↓	CSIII vs. C CSIV vs. C
202723_s_at	*FOXO1*	3.3 × 10^−4^	−2.29 ↓	−1.31 ↓	−1.21 ↓	−1.42 ↓	CSI vs. C
201383_s_at	*NBR1*	8.8 × 10^−4^	−1.67 ↓	−1.52 ↓	−1.86 ↓	−1.98 ↓	
200804_at	*TMBIM6*	2.8 × 10^−4^	−1.46 ↓	−1.15 ↓	−1.50 ↓	−1.90 ↓	CSIV vs. C
201178_at	*FBXO7*	4.6 × 10^−5^	−1.14 ↓	−1.09 ↓	−1.35 ↓	−1.53 ↓	CSIII vs. C CSIV vs. C
200633_at	*UBB*	7.4 × 10^−3^	1.03 ↑	−1.25 ↓	−1.25 ↓	−1.13 ↓	CSI vs. C
208980_s_at	*UBC*	1.9 × 10^−3^	−1.51 ↓	−1.42 ↓	−1.38 ↓	−1.47 ↓	

Legend: *BECN1—Beclin1; FBXO7—F-box only protein 7; HSP8—Heat Shock Protein Family A (Hsp70) Member 8; MFN1—Mitofusin-1; NBR1—Mitofusin-1; *PTPN22*—protein tyrosine phosphatase non-receptor type 22; TMBIM6—Transmembrane BAX Inhibitor Motif-6; UBC—ubiquitin C*.

## Data Availability

The data presented in this study are available upon request from the corresponding author.

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
