# Peer review of "Relationship between the Ubiquitin–Proteasome System and Autophagy in Colorectal Cancer Tissue"

_biomedicines, 2023, doi:10.3390/biomedicines11113011_

Round 1
Reviewer 1 Report
Comments and Suggestions for Authors
In this study, authors focused on exploring the changes in the transcriptional activity of genes involved in the autophagy and ubiquitination process (UPS) in colorectal cancer tissues. Overall, to explore the changes of genes involved in protein degradation systems in CRC has interest and merit. However, the study design is somehow illogical. In addition, the results were preliminary and not well supported the conclusions. Furthermore, the current manuscript requires appropriate improvements. Several representative concerns are provided below.
1. Abbreviations should be well defined, e.g. CSI and CSII (#22) were not defined.
2. Some descriptions are confusing, e.g. “to the autophagy-lysosomal system (AL), UPS has a high selectivity because only ubiquitin-labeled proteins are excluded”(#45-46)
3. #49-50 “Authors indicates that the autophagy receptor p62/sequestosome-1 as the principal molecule that regulates the cross-talk between the two systems [ 5-7].”, References 5-7 do not appear to show that SQSTM1 is a regulator of either system.
4. Legend in #113-116, there is no color presented in Table 3.
5. In Table 4, most mRNA levels in each CSn were compared with control; however, only HSPA8 mRNA levels were compared between CSI vs CSIV and CSII vs CSIII. The reason should be addressed.
6. Since this study does not provide sufficient and conclusive evidence, the discussion needs to be restrained, and the limitations should be well described and discussed.
Comments on the Quality of English LanguageIt needs appropiate improvement.
Reviewer 2 Report
Comments and Suggestions for Authors
biomedicines-2630948
Title: Relationship between the ubiquitin–proteasome system and autophagy in colorectal cancer tissue
Authors: Martyna Bednarczyk, Małgorzata Muc-Wierzgoń, Sylwia Dzięgielewska-Gęsiak and Dariusz Waniczek
In general, the data in present studies are good and support the major conclusions of this manuscript. However, following issues need to be considered prior to considering the manuscript of publication.
Specific comments are as follows:
[Major concern]
1. Notation of genes and proteins: In this paper, there are many occurrences of descriptions pertaining to several genes and their corresponding proteins. However, the gene nomenclature (commonly represented in italics) is not consistently adhered to, causing significant confusion. Furthermore, it has been asserted that five key genes hold significant importance based on experimental results. To alleviate confusion, it is crucial to consistently list these five genes in a clear manner, starting with their presentation in the abstract. Additionally, when providing data related to these five genes, it is essential to arrange the information in a consistent and predefined order. By doing so, readers will easily comprehend the content.
2. Abbreviations: Most journals require that an abbreviation be spelled out at its first occurrence in the text, followed by the abbreviation in parentheses. (Exception: If the abbreviation is on the journal's list of permitted abbreviations, this need not be done.) Thereafter, only the abbreviation may be used. Note also that abbreviations need to be independently defined in the abstract and the main text of the paper. Abbreviations need not be introduced if they are not used again.
When using abbreviations, it is important to always start with the full name followed by the abbreviation enclosed in parentheses. Many instances of authors doing the opposite have been observed, so please correct all of them.
Some abbreviations have been mentioned earlier, but they are being repeated here. Therefore, it would be beneficial to systematically review and clarify the abbreviation usage from the beginning.
3. English: In general, the English sentences are well-constructed, but there are several instances where they lack smoothness, and numerous typos have been identified. It would be advisable to seek the assistance of an expert to correct the English sentences.
4. When abbreviating Clinical Stages (CS) I-IV, various different patterns are being used. Please always employ a consistent pattern. Examples: CS I vs. CSI vs. CS_I, etc.
5. In cases where abbreviations are used within figures or tables, please list these abbreviations along with their corresponding full names in the figure legends or at the bottom of corresponding tables. If there are two or more abbreviations, arrange them in alphabetical order.
6. Materials and Methods section - When naming a particular chemical company, you must provide location information such as company name, city and/or state (abbreviation in the USA and Canada) and country. Once you have named a company with the information, you should only mention a company’s name thereafter. Information about several companies is wrong, so check and correct it. It is generally well written in this paper, but there are a few mistakes, so find them and correct them.
[Minor concern]
1. Line 19: ‘HG – U133A’ should be written as ‘HG-U133A’ according to the manual from the company.
2. Line 108: Table2 and Table3 should be written as Table 2 and Table 3.
3. Line 193: The authors have described this paper as if it were the work of a third party, even though it is their own research. Please verify and correct such instances in the text.
4. Line 252: Correct the cited references for [Rehman, Zhao].
5. Line 280: HSPA 8 should be written as HSPA8.
Overall, the manuscript can be considered to publication after major revision as indicated above.
Comments on the Quality of English Languagebiomedicines-2630948
Title: Relationship between the ubiquitin–proteasome system and autophagy in colorectal cancer tissue
Authors: Martyna Bednarczyk, Małgorzata Muc-Wierzgoń, Sylwia Dzięgielewska-Gęsiak and Dariusz Waniczek
In general, the data in present studies are good and support the major conclusions of this manuscript. However, following issues need to be considered prior to considering the manuscript of publication.
Specific comments are as follows:
[Major concern]
1. Notation of genes and proteins: In this paper, there are many occurrences of descriptions pertaining to several genes and their corresponding proteins. However, the gene nomenclature (commonly represented in italics) is not consistently adhered to, causing significant confusion. Furthermore, it has been asserted that five key genes hold significant importance based on experimental results. To alleviate confusion, it is crucial to consistently list these five genes in a clear manner, starting with their presentation in the abstract. Additionally, when providing data related to these five genes, it is essential to arrange the information in a consistent and predefined order. By doing so, readers will easily comprehend the content.
2. Abbreviations: Most journals require that an abbreviation be spelled out at its first occurrence in the text, followed by the abbreviation in parentheses. (Exception: If the abbreviation is on the journal's list of permitted abbreviations, this need not be done.) Thereafter, only the abbreviation may be used. Note also that abbreviations need to be independently defined in the abstract and the main text of the paper. Abbreviations need not be introduced if they are not used again.
When using abbreviations, it is important to always start with the full name followed by the abbreviation enclosed in parentheses. Many instances of authors doing the opposite have been observed, so please correct all of them.
Some abbreviations have been mentioned earlier, but they are being repeated here. Therefore, it would be beneficial to systematically review and clarify the abbreviation usage from the beginning.
3. English: In general, the English sentences are well-constructed, but there are several instances where they lack smoothness, and numerous typos have been identified. It would be advisable to seek the assistance of an expert to correct the English sentences.
4. When abbreviating Clinical Stages (CS) I-IV, various different patterns are being used. Please always employ a consistent pattern. Examples: CS I vs. CSI vs. CS_I, etc.
5. In cases where abbreviations are used within figures or tables, please list these abbreviations along with their corresponding full names in the figure legends or at the bottom of corresponding tables. If there are two or more abbreviations, arrange them in alphabetical order.
6. Materials and Methods section - When naming a particular chemical company, you must provide location information such as company name, city and/or state (abbreviation in the USA and Canada) and country. Once you have named a company with the information, you should only mention a company’s name thereafter. Information about several companies is wrong, so check and correct it. It is generally well written in this paper, but there are a few mistakes, so find them and correct them.
[Minor concern]
1. Line 19: ‘HG – U133A’ should be written as ‘HG-U133A’ according to the manual from the company.
2. Line 108: Table2 and Table3 should be written as Table 2 and Table 3.
3. Line 193: The authors have described this paper as if it were the work of a third party, even though it is their own research. Please verify and correct such instances in the text.
4. Line 252: Correct the cited references for [Rehman, Zhao].
5. Line 280: HSPA 8 should be written as HSPA8.
Overall, the manuscript can be considered to publication after major revision as indicated above.
Round 2
Reviewer 1 Report
Comments and Suggestions for Authors
The previous issues have been addressed, and the manuscript is also properly revised. No further concerns have arisen.
Reviewer 2 Report
Comments and Suggestions for Authors
Accept in present form.
However, several minor typos should be corrected during the proofreading at the office.
Comments on the Quality of English LanguageAccept in present form.
However, several minor typos should be corrected during the proofreading at the office.